# Cutting-Edge Platforms for Analysis of Immune Cells in the Hepatic Microenvironment—Focus on Tumor-Associated Macrophages in Hepatocellular Carcinoma

**DOI:** 10.3390/cancers14081861

**Published:** 2022-04-07

**Authors:** Daniel E. Millian, Omar A. Saldarriaga, Timothy Wanninger, Jared K. Burks, Yousef N. Rafati, Joseph Gosnell, Heather L. Stevenson

**Affiliations:** 1Department of Pathology, University of Texas Medical Branch, Galveston, TX 77555, USA; damillia@utmb.edu (D.E.M.); omsaldar@utmb.edu (O.A.S.); jogosnel@utmb.edu (J.G.); 2Department of Microbiology and Immunology, University of Texas Medical Branch, Galveston, TX 77555, USA; tiwanning@utmb.edu; 3Department of Leukemia, University of Texas MD Anderson Cancer Center, Houston, TX 77030, USA; jburks@mdanderson.org; 4School of Medicine, University of Texas Medical Branch, Galveston, TX 77555, USA; ynrafati@utmb.edu

**Keywords:** cancer, CyTOF, HCC, liver, macrophages, spectral imaging, hepatic microenvironment, TAMs, TME, spatial genomics, scRNA-seq

## Abstract

**Simple Summary:**

Hepatocellular carcinoma is the most common primary liver malignancy in the United States. Macrophages are immune cells that play a critical role in the promotion of cancer growth and configuration of the hepatic microenvironment. Studying intrahepatic macrophages is challenging because they are difficult to isolate, they transform their phenotype upon manipulation, and in vivo animal models poorly replicate the liver microenvironment. Understanding the complexity of intrahepatic macrophage populations is crucial because they coordinate antitumoral immunity. Application of novel methods that can detect immune cell phenotypes, along with their spatial co-localization in situ is critical and timely.

**Abstract:**

The role of tumor-associated macrophages (TAMs) in the pathogenesis of hepatocellular carcinoma (HCC) is poorly understood. Most studies rely on platforms that remove intrahepatic macrophages from the microenvironment prior to evaluation. Cell isolation causes activation and phenotypic changes that may not represent their actual biology and function in situ. State-of-the-art methods provides new strategies to study TAMs without losing the context of tissue architecture and spatial relationship with neighboring cells. These technologies, such as multispectral imaging (e.g., Vectra Polaris), mass cytometry by time-of-flight (e.g., Fluidigm CyTOF), cycling of fluorochromes (e.g., Akoya Biosciences CODEX/PhenoCycler-Fusion, Bruker Canopy, Lunaphore Comet, and CyCIF) and digital spatial profiling or transcriptomics (e.g., GeoMx or Visium, Vizgen Merscope) are being utilized to accurately assess the complex cellular network within the tissue microenvironment. In cancer research, these platforms enable characterization of immune cell phenotypes and expression of potential therapeutic targets, such as PDL-1 and CTLA-4. Newer spatial profiling platforms allow for detection of numerous protein targets, in combination with whole transcriptome analysis, in a single liver biopsy tissue section. Macrophages can also be specifically targeted and analyzed, enabling quantification of both protein and gene expression within specific cell phenotypes, including TAMs. This review describes the workflow of each platform, summarizes recent research using these approaches, and explains the advantages and limitations of each.

## 1. Introduction

Hepatocellular carcinoma (HCC) is the most common primary liver malignancy in the United States and worldwide [1]. The mortality rate from HCC is increasing faster than any other cancer in the United States and is also a leading cause of cancer deaths globally, accounting for more than 700,000 deaths each year [2]. So far, the implementation of new diagnostic strategies and therapeutic regimens has not yet resulted in a significant reduction in mortality, as HCC is the second most lethal tumor, outranked only by pancreatic cancer, in the United States [2]. The hepatic microenvironment in HCC is composed of a heterogenous population of cells with distinct genetic and phenotypic properties and various effects on tumor progression. Tumor-associated macrophages (TAMs) are a critical subpopulation that can promote tumor growth within the hepatic microenvironment and unique phenotypes with diverse functional properties have been described [3,4].

The specific role of TAMs in promotion of HCC is controversial and poorly understood since most studies to date rely upon traditional in vitro and in vivo animal models (e.g., mouse models) that poorly replicate the tumor microenvironment (TME) in humans [5,6,7]. The prevailing view is that TAMs in HCC are involved in immunosuppression, angiogenesis, epithelial-mesenchymal transition, cytokine secretion, enhancement of metastasis, and prolongation of stemness [4]. Preclinical studies have found a strong correlation between macrophages infiltrating the TME and poor prognosis [4,6,7]. The profound influence of TAMs on tumor progression is confirmed by a growing body of data indicating that agents targeting TAMs are critical for optimal HCC therapy [8]. Despite various techniques for isolating and characterizing TAMs, their study is challenging because of the following: (1) isolation of the cells from human liver tissue is laborious, (2) macrophages transform their phenotype upon manipulation, and (3) in vitro and mouse model systems poorly replicate the TME, and chronic disease observed in humans [9]. Therefore, other methods that can study the complex hepatic microenvironment in situ in human liver tissue may be better approaches. TAMs are a promising target for immunotherapy, and accurate characterization will be necessary for successful implementation of precision medicine approaches.

In this review, emerging platforms that can characterize intrahepatic macrophages in situ in human liver tissue will be discussed. The basic principles of each technique are outlined, along with the advantages and disadvantages of each platform. We also suggest how each may contribute to the development of TAM-targeting drugs and personalized medicine. Finally, recent investigations of TAMs in HCC progression are described. 

## 2. Multispectral Imaging

### 2.1. Traditional Methods for Phenotyping Immune Cells In Situ

For more than 100 years, light microscopy has been the traditional method for pathologists to evaluate tissue specimens. The information obtained by reviewing hematoxylin and eosin-stained tissue slides with brightfield microscopy is limited. With the development of immunotherapy and increased implementation of precision medicine approaches, pathologists need to provide more than just a diagnosis or margin status. They should also report information about prognostic factors and expression of therapy-related targets, such as programmed death-protein 1 (PD-1) and its ligand (PDL-1), whenever possible [10,11,12]. Although development of chromogenic immunohistochemistry (IHC) in diagnostic and research pathology was a major advancement, its use is limited since it cannot characterize multiple immune cell phenotypes, particularly when multiple antigens are colocalized on the same cell type or same cellular compartment (e.g., characterization of TAMs) [11,13,14]. Macrophage markers can be stained on individual IHC slides and then quantified. However, newer techniques that use multiplex immunofluorescence (IF) combined with advanced imaging software for data analysis are more accurate and feasible for phenotyping studies [14,15].

### 2.2. Multiplex Immunofluorescen Staining Technique Followed by Multispectral Imaging Analysis

Multispectral imaging is a technique that overcomes the above challenges since it is equipped with a specialized camera. In multispectral imaging (MSI), antibodies are conjugated with tyramide-fluorophores, called tyramide signal amplification (TSA). These types of fluorophores are suitable for MSI because the tyramide-fluorophores attach covalently to tyrosine amino acids; each of the antibodies are then removed sequentially, leaving the stain intact when the antibody probes are removed. The signal each fluorophore emits at specific wavelength is unmixed using a spectral library and a microscope that has a paired of excitation/emission filter sets specific to the emission spectrum of each fluorophore employed.

MSI is ideal for analyzing multiple immune cell phenotypes in situ since it preserves the tissue architecture and can determine the spatial location of cells within a region of interest or whole scanned slide. MSI with platforms such as the Vectra or Polaris (Akoya Biosciences, Marlborough, MA, USA) can detect up to 6-8 biomarkers in a single formalin-fixed paraffin-embedded (FFPE) tissue section (Figure 1) [15]. The staining method is similar amongst most of the MSI technologies discussed in this review, as the process involves the following steps: (1) deparaffinization of the tissue slide, (2) slide fixation, (3) antigen retrieval, (4) addition of a blocking reagent to prevent non-specific binding of antibodies to the tissue or to Fc receptors, (5) incubation and binding of an unmodified primary antibody to its target, (6) horseradish peroxidase (HRP)-conjugated secondary antibody incubation and binding of the HRP-conjugated antibody to the primary antibody, (7) HRP enzyme-mediated in situ deposition of the tyramide-fluorophore that covalently binds to the tissue near the target, (8) antibody removal, (9) counterstaining of the tissue with 4,6-diamidino-2-phenylindole (DAPI) or other nuclear stain, and (10) mounting of the slide and visualization using fluorescence microscopy (Figure 1). Note that after step 8, steps 3 to 8 are repeated until the desired number of antibodies have been used. These steps are unique to MSI that uses TSA. 

MSI can be conducted with IF or brightfield chromogens; however, use of light microscopy is limited to fewer channels than IF as well as by broad spectral absorption. A recent study was able to overcome this limitation by using new chromogens with narrow spectral absorption and matched illumination channels [10]. Another study used chromogenic multiplexed IHC on a prostate cancer tissue microarray cohort consisting of 462 duplicate cores with outcome data and 384 duplicate cores from different disease stages. The authors observed that chromogenic multiplexed IHC, in combination with digital biomarker analysis software, was a reliable alternative for objective and high-throughput biomarker quantification and colocalization [16]. Further development of this technique may accelerate the integration of MSI into general pathology practice [3,17].

### 2.3. Vectra 3, Vectra Polaris and CODEX

In the field of phenoptics digital pathology, the Vectra 3 is a well-known example of MSI technology. The Vectra 3 is an automated quantitative pathology imaging system, with either a 6-slide capacity or 200 slide hotel attached to the scanner, that detects up to 35 lambda channels (wavelengths) in the same tissue section, which can be used to separate and identify 9 fluorochromes (8 antibodies and DAPI, nuclear counterstain). While the system is capable of handling 9 fluorochromes, it is more commonly used to detect 7 fluorochromes due to complexity. The platform was designed to operate with the inForm imaging analysis software version 2.4.0. The Vectra 3 acquires images from tissue sections labeled with Opal-conjugated fluorophores combined with TSA, which enhances the intensity of the fluorophore signal [18]. Then, the Vectra 3 multispectral camera is programed to capture images every 20 nm across the spectrum of 420 nm to 720 nm from various excitation filters resulting in the aforementioned 35 lambda channels collection. The extraction of the spectral data from the images is possible because the Vectra 3 uses a liquid crystal tunable filter (optical filter) that operates by transmitting a narrow band of wavelengths. The optical filter is key because viewing a spectral image requires capturing a collection of images of different wavelengths at each pixel location. 

After gathering a complete spectrum of the image at every pixel location, the inForm (Akoya Biosciences) software analyzes the tissue images using the following workflow: (1) unmixing of each multispectral signal allows the identification and separation of weak and overlapping signals from background autofluorescence, (2) artificial intelligence algorithms are trained to automatically detect and segment specific tissue types, (3) after being trained, the software will automatically locate and analyze specific regions within an image or across images, (4) machine-learning algorithms separate cell phenotypes in a tissue section, (5) a positive threshold is set based on the quantification of the stain intensity and the calculated H-score, and (6) large sets of images or slides can be batch-processed, reviewed, and merged to hasten and improve image analysis efficiency.

The Vectra Polaris is an upgraded version of the Vectra 3 that allows a higher throughput, improved image quality, brightfield and/or IF whole slide scanning, and superior quantitative analysis. The resolution of the Vectra Polaris at higher magnification is also improved with the values of: (1) 40× = 0.25 μm/pixel, (2) 20× = 0.5 µm/pixel, and (3) 10× = 1.0 µm/pixel. The Polaris also features bandpass filters optimized to a 7-fluorophore assay in the second generation of Opal featuring two new fluorochromes Opals 480 and 780, replacing Opals 540 and 650. By replacing these two, fluorophores bandpass filters were made possible, thereby reducing the lambda collections to nine channels, but still retaining the multispectral benefits. The largest benefit is that data collection across the entire slide is now possible. 

For comparison, cyclic methods, including Akoya CODEX, CycIF, Lunaphore Comet, Miltenyi MACSIma, Leica Microsystems Cell Dive, and Brukers Canopy, to name a few, cycle fluorophores on and off the antibodies serially and align the images collected to create a higher-plex image, depending on the number of cycles employed [4,19,20,21]. CODEX employs an oligo cleavage, Canopy quenches the signal leaving the antibody and probe, Miltenyi MACSIma erases the fluorescence signal of samples that have been stained with fluorochrome-conjugated antibodies via photobleaching, and others methods remove everything. These methods capture individual cycles rapidly, but this step must be repeated multiple times, and the user faced with the daunting challenge of aligning all the images collected per cycle. This dramatically affects the throughput as research interests scale to higher parameter numbers. However, the RareCyte Orion platform generates same-day whole-slide images with sub-cellular imaging resolution in a single stain, single scan workflow. This platform scans up to 21 fluorescence channels simultaneously decreasing the turnaround time as it does not use an iterative or cyclic process. Antigenic preservation of the epitopes that the antibodies identify is optimized because only one staining cycle is performed. The Leica Microsystems Cell Dive uses a protocol that reconstructs a component image by layering multiple images generated through staining cycles, allowing complex multiplexing of the tissue at single-cell resolution.

The CODEX platform is a modification of the conventional multiplex technique in which antibodies are tagged with unique oligonucleotides and dye-oligonucleotides that function as barcodes and reporters, respectively. CODEX provides high-parameter IF imaging of fresh-frozen and FFPE tissue because the barcodes (oligonucleotides) and reporters (dye-oligonucleotides) are iteratively hybridized and dehybridized across multiple cycles. An advantage of Codex is its capacity to probe a broad range of molecules of interest in a single tissue slide. The CODEX workflow and chemistry was recently applied in a colon cancer study, in which the configuration, cell to cell interaction, and spatial organization of the TME was characterized in great detail [22]. In the study, the investigators profiled 56 protein markers across 140 regions of interest from 35 advanced-stage colorectal cancer patients. They correlated an enriched association between unique immune cell neighborhoods and outcomes. Enriched populations of PD-1^+^ CD4^+^ T cells within a granulocyte cellular neighborhood correlated with better survival. In contrast, a worse outcome was associated with an enriched cellular neighborhood where tumor and immune cells where coupled, where T cells and macrophages where fragmented, and communication was disrupted. They proposed that using their approach on a larger cohort may yield clinically relevant biomarkers, treatment regimens, and better comprehension of the biological landscape where immune cells coordinate antitumoral immunity. 

### 2.4. Phenotyping Tumor-Associated Macrophages in the Tumor Microenvironment Using Multispectral Imaging

For the study of macrophages by MSI microscopy, the addition of more channels and narrow spectral absorption are critical since TAMs are often positive for multiple markers in the same cellular compartment and each phenotype may have unique biological functions [3]. TAM phenotype, polarization, and degree of infiltration have been correlated with clinical prognosis, tumor behavior, and response to therapy [4,23,24]. For example, using MSI on tissue biopsies obtained from 66 patients with HCC revealed that high co-expression of CD38^+^ CD68^+^ on TAMs was associated with a better prognosis after surgery, in comparison to patients whose tumor only expressed a high density of CD68^+^ [25]. In addition, CD38^+^ expression was found to be enriched in TAMs with a cytokine profile similar to that of M1 macrophages. 

Initiation of TAM-targeted therapy is another instance where determination of the presence of specific macrophage phenotypes is crucial for effective treatment. Some authors propose that TAM-targeted therapy should be divided into four categories for HCC: (1) inhibiting recruitment of monocytes, (2) eliminating TAMs already present in tumor tissue, (3) re-educating the functions of polarized TAMs by making M2 macrophages more M1-like, and (4) neutralizing the tumor-promoting factors of TAMs [4]. A limitation of these approaches is that to be effective, the specific target must be present. Standard IHC has not proven efficient for detecting the presence or absence of multiple TAM targets because it requires multiple slide levels, and tissue biopsies are scarce in most cases. MSI makes this approach much more feasible by enabling the detection of multiple markers in a single unstained tissue slide. For example, Saldarriaga et al. [18] identified numerous phenotypes of intrahepatic macrophages in different types of chronic liver diseases using only a single unstained slide from each patient’s liver biopsy.

### 2.5. Limitations of Multispectral Imaging and Future Directions

Studies using MSI in cancer research are growing in number. The field of immuno-oncology has had several breakthroughs with the implementation of immunotherapies, such as the immune checkpoint inhibitor family (e.g., nivolumab, ipilimumab, and pembrolizumab). Immunophenotyping can be correlated with prognosis and response to immunotherapy. A major challenge is that it is often difficult to predict who will respond to each therapy. Characterization of the various cell phenotypes present in the complex hepatic microenvironment will facilitate personalized approaches to therapy.

An obstacle to integrating MSI into the clinical context is the complexity and lack of clinical data [26]. The translation of MSI into patient care is dependent on standardizing a workflow validated in multiple centers, clinical trials, and clinical laboratories [27]. There is also resistance from pathologists to transition from brightfield to other types of microscopies. Quality control is another issue, as standardizing reproducible antibody performance for MSI in the histopathology laboratory has not been straightforward [11]. These challenges may soon be overcome due to increasing expertise in computerized-aided software and the use of machine learning and artificial intelligence. The development of spectral libraries using advanced imaging software platforms, such as inForm or Visiopharm (Hoersholm, Denmark), has solved these technical problems by improving the user interface and providing automated image analysis algorithms. 

When these limitations are resolved and MSI is more well-adapted by pathologists and histopathology laboratories, it will play a central role in the diagnosis and management of many tumors, including HCC [11,26,28]. In clinical oncology, trials using MSI have shown that the presence of specific receptors on tumor infiltrating lymphocytes, not only on TAMs, are key mediators of tumor progression [29]. Ideally, these will allow personalization of treatment and the ability to better characterize the TME [28,30,31]. Complex spatial and nearest neighbor analyses, using platforms such as CODEX, highlight the potential of MSI in future clinical studies and translational research.

## 3. Cytometry by Time-of-Flight

### 3.1. Cytometry by Time-of-Flight: A Fusion of Multiple Techniques

It was in the late 1960s that the roots of fluorescence-activated cell sorting (FACS) originated [32,33]. Some of the core technological principles used in the analysis and sorting of FACS remain unchanged [32]. FACS is a technique that involves staining cells with fluorescently labeled antibodies directed against a cell surface or intracellular epitope. Then, the labeled cells are passed through an excitation laser [34]. The fluorophores are excited by the laser beam and emit light at a certain wavelength. As each cell in the suspension passes in single file, the wavelength of the light emitted by each stained cell is capture by a photoelectric cell that gauges the intensity of the fluorescent excitation. These measurements are plotted as a distribution of emitted light signals. Flow cytometry follows the same principle as FACS, but it does not retain a purified population of cells. Methods using elemental mass spectrometry which label cells with antibodies conjugated with isotopes of different atomic weights rather than fluorescent molecules have gained popularity in both the medical and research fields [35]. Flow cytometry is a standard method for the diagnosis of leukemias and lymphomas in clinical laboratories and remains the main method for characterizing immune cells in the field of immunology. However, mass cytometry is growing in demand in cancer and immunology research.

### 3.2. Cytometry by Time-of-Flight Workflow

Understanding the principles of flow cytometry, FACS and elemental mass spectrometry is necessary before designing studies using cytometry by time-of-flight (CyTOF) because this platform fuses certain aspects of each of these techniques [35,36,37]. CyTOF is unique among other technologies because it can analyze single cells in suspension, similar to flow cytometry. Additionally, analogous to a high-plex MSI, it has been adapted to probe paraffin-embedded tissue sections in situ, using a variation called imaging mass cytometry (IMC; Fluidigm, South San Francisco, CA, USA). CyTOF was developed to expand the number of cellular parameters that could be measured simultaneously. This platform provides a novel solution to the fluorophore-labeled techniques, such as FACS and flow cytometry, in which spectral overlap between fluorescence emission profiles limits the number of markers that can be used. Furthermore, it takes advantage of the ability of elemental mass spectrometry (Inductively coupled plasma mass spectrometry or ICP-MS) to distinguish isotopes with high accuracy.

CyTOF replaces the fluorophores by using element isotopes as labels. The isotope reporter is above 89 amu, and includes the transition metals, but also the rare earth/lanthanides, actinides, and others. An elegant advantage of this substitution is that, with non-biological markers, such as metal-isotopes (e.g., lanthanide), the tissue background noise can be removed since it is clear that the signal is coming from the non-biological marker [38]. The cells and/or tissue with an epitope-specific antibody conjugated to an element isotope reporter are vaporized and quantified by a time-of-flight (TOF) mass spectrometer (Figure 2A) [34,36]. In suspension CyTOF, each cell is nebulized into single-cell droplets and passed through the plasma torch that ionizes the sample. Then, the isotope reporters are quantified through the TOF chamber. The data can be presented in a similar fashion to that of conventional flow cytometry plots and heat maps of induced phosphorylation [37]. This platform can detect more than 50 labels with a theoretical limit of 135 channels in the current instrument configuration [38]. This is substantially more than what is detected by conventional flow cytometry. For example, the Cytek Aurora, a multispectral flow cytometer has demonstrated the ability to detect 40 channels successfully [39]. While the suspension CyTOF can detect more than 50 labels currently, the imaging CyTOF is more limited and has collected sample sets of 42 antigens in a single tissue [35,37,40]. This variance is due to some isotopes natively adhering to glass, creating complications in their use for imaging approaches where the tissue is housed on a glass slide. 

The IMC workflow follows the same underlying principles as suspension CyTOF, however, since a laser ablates the tissue during the analysis and the architecture needs to be reconstructed. As shown in Figure 2B, in IMC, the first step is to label a tissue section with the pre-selected panel of antibodies conjugated with stable isotopes. Then, the tissue is inserted into an ablation chamber and a camera acquires images from the tissue slide for region of interest (ROI) selection. In the ablation chamber, ROIs are scanned by a solid laser operating at 213 nm and directed to 1 µm area per ablation spot, which equates to the pixel resolution of the resulting image. On each laser shot, the laser beam ablates a spot of tissue stained with the metal-tagged antibodies, forming aerosol plumes. The plumes of tissue matter are directed by a flow of inert gas (Argon) into a coupling tube that delivers them to an inductively coupled plasma ion source where they are vaporized, atomized, and ionized. Then, high-pass ion optics remove the low-mass ions (Carbon, Oxygen, Nitrogen, etc.) that are not of interest before the metal-isotope ion cloud passes into the TOF mass spectrometer for analysis of isotope abundance, and, therefore, epitope abundance. The number of metal-tagged antibodies from each tissue spot, corresponding to a laser shot, are counted simultaneously, and mapped to the location of each spot. A useful approach to understand how IMC indexes each tissue spot, is to imagine that each laser shot represents one pixel of the tissue image, whilst the isotopes are the colors (RGB) of each pixel. The spatial resolution of the tissue image can be optimized by decreasing the speed or size of the laser shot.

### 3.3. Use of Cytometry by Time-of-Flight for Characterization of Multiple Macrophage Phenotypes in the Hepatic Microenvironment

In a study dissecting the cancer-immune landscape in HCC using CyTOF, it was observed that TAMs play an essential role in the configuration of an immunosuppressive gradient across the tumor, non-tumor, and peripheral blood microenvironments [41]. Interestingly, the study also identified a chemotactic gradient in the tissue microenvironment, composed of the chemokines CCL20 and CXCL10, attracting TAMs and resident natural killer (NK) cells. It was observed that both types of cells induced an immunosuppressive microenvironment via the expression of high levels of IL-10 by TAMs and low levels of granzyme B by NK cells. Furthermore, this study showed that HBV patients who developed HCC had increased expression of both IL-10 by TAMs and the exhaustion markers PD-1, CD152, and Lag-3 by tumor-infiltrating lymphocytes, as compared to non-tumor-infiltrating lymphocytes. The authors claimed that this observation suggests that HCC patients treated with either macrophage-targeted therapy or immune checkpoint inhibitors will not have an illicit off-target immune response in the remaining non-cancerous tissue infected with HBV [41]. What appears certain from this study, and others, is that targeting the TAM chemokine axis can enhance the effect of immune checkpoint inhibitors by recruiting and activating more CD8^+^ and CD4^+^ T cells, which increases the number of tumor-infiltrating lymphocytes and makes the microenvironment less immunosuppressive [42,43,44,45].

The topology of the tissue microenvironment was analyzed by Sheng et al. [24] using IMC since suspension CyTOF cannot adequately assess intratumoral heterogeneity in situ, in the context of intact architecture or spatial context [24,46]. IMC provides in situ information regarding the tissue architecture and neighboring cell interactions by limiting the tissue analysis to ROIs. The topological analysis of Sheng et al. [24] revealed three important observations. First, the tissue microenvironment of HCC is divided into regional cellular functional units that impact the prognosis of patients. Second, CD80^+^/86^+^ infiltrating macrophages activated CD8^+^ T cells via the CD28 signaling pathway. In contrast to CD80^+^/86^+^ infiltrating macrophages, Kupffer cells positive for the PD-L1 marker inactivated CD8^+^ T cells by dephosphorylating CD28. Third, in a mouse model, depletion of Kupffer cells augmented the number of infiltrating macrophages, reduced the tumor growth, and improved the efficacy of an anti-PD-1 inhibitor. 

### 3.4. Cytometry by Time-of-Flight Limitations and Future Directions

This technology is powerful and enables profiling of many cellular features, with an enhanced resolution of TAM phenotypes, and provides the opportunity for high-dimensional analyses [35]. As discussed above, the results from experiments using CyTOF have raised the issue that the optimal use of new immunomodulators should include technologies that stratify patients into responders and non-responders [47]. The justification for this argument is that CyTOF played an essential role in generating empirical data suggesting that the presence of PD-L1+ Kupffer cells will reduce the efficacy of an anti-PD-1 inhibitor. Limitations of CyTOF include: (1) vaporizing cells is an irreversible step, (2) during IMC, laser ablation results in loss of the tissue for further testing, and (3) the sensitivity for measuring lower expressing molecular characteristics is challenging due to the directly conjugated antibody (no secondary amplification), [35,48]. The requirement for large quantities of cells to be analyzed may be a hinderance for studying TAMs, if they are present in low numbers in a tissue specimen. Despite these limitations, studies using CyTOF have provided new insights into the role of TAMs in the tissue microenvironment, HCC progression and prognosis, and response to therapy. Therefore, highly multiplexed epitope-based imaging may become a cornerstone in the study of TAMs and HCC as biomedical research translates spatial genomic data into more precise medical treatments.

## 4. Digital Spatial Profiling

### 4.1. Digital Spatial Profiling: From DNA Microarrays to Spatial Genomics

DNA microarrays use a pre-selected collection of DNA probes assembled on a solid surface to quantify the presence of a complementary sequence of DNA in a sample [49]. This technology emerged from the need to rapidly screen thousands of *Escherichia coli* colonies to identify clones containing DNA that was complementary to a probe [50]. Real time polymerase chain reaction (PCR) is another technology that quantifies gene expression by measuring the amplification of a target sequence during PCR rather than at the end of the reaction. Both techniques quantify gene expression with high precision but lack spatial information of protein and gene expression. Characterization of the tissue microenvironment by profiling the spatial relationships of protein and gene expression is critical for understanding the role of TAMs in HCC. 

Spatial characterization of whole transcriptome analysis (WTA) and protein expression in TAMs requires two complementary features. First, obtaining new insights into the role TAMs play in the microenvironment requires detailed profiling of WTA and protein expression. Spatial characterization of both genomic and protein expression in TAMs will provide compelling evidence of the crosstalk between macrophages and other immune cells, such as CD8^+^ and NK cells, in the tissue microenvironment that control progression of malignant cells. Second, this must be accomplished while simultaneously retaining the hepatic architecture. Nanostring developed a technology, called GeoMx digital spatial profiling (GeoMx DSP), which addresses this critical need. This unique platform is able to perform WTA and detect protein expression within a ROI containing multiple to hundreds of cells, depending on ROI size, from a single FFPE tissue slide. Visium Spatial Gene Expression is another platform that can be used to address the need mentioned above using a different workflow that is explained below.

### 4.2. Digital Spatial Profiling Workflow

DSP is a technology that combines multiplexed, spatial characterization of pre-selected proteins and/or RNA probes in a tissue by detecting oligonucleotide barcodes conjugated by photocleavable linker to either primary antibodies or nucleic acid probes (Figure 3A). The probes target a transcript or protein of interest. A feature of the probes is the unique barcode RNA sequence that allows indexing of individual proteins or mRNA. After the target-specific barcodes are liberated by the UV laser from the selected ROIs, they are then counted by an nCounter platform (Figure 3B). GeoMx DSP stores data from each ROI after the expression of a target is quantified. This tissue based spatial characterization of genes and proteins is possible since this platform’s UV laser is guided to work in one region of interest at a time and each target quantified by the nCounter is mapped back to the ROI from where it originates. This technology provides significant new insights into the role that TAMs play in the progression of HCC by combining multiplex microscopy and spatial genomics. It is capable of profiling the cell phenotype and corresponding gene expression in a single ROI. 

Another similar platform is Visium Spatial Gene Expression, a spatial transcriptomics platform that works with cell capture slides that contain four capture areas with 5000 barcoded spots. Multiple barcoded spots capture oligonucleotides that bind to the RNA in the tissue. Each barcoded spot captures the transcripts from 1–10 cells. Thus, Visium Spatial Gene Expression is approaching single-cell level, however, it can provide high-resolution transcriptomics data while keeping the liver architecture intact. Briefly, tissue sections are placed onto the capture areas, stained with H&E or IF staining, and imaged. Then, the tissue is permeabilized to release the mRNA so it can be captured, and the resulting mRNA is synthesized into cDNA. Sequencing libraries are then prepared. 

The Rebus Esper Spatial Omics and Nanostring CosMx are unique DSP platforms as they provide spatial transcriptomics at single-cell resolution while maintaining the architectural context across large tissue sections. The Rebus Esper can generate spatially resolved quantitative multiplexed data for more than 100,000 cells in less than two days by combining high resolution images, optimized assays, and using software with advanced data processing [51]. Like other DSP platforms, ROIs are selected from which spatial data is acquired. The automated system performs all reactions, washes, and imaging acquisition. Once the images are acquired, they are processed by cutting-edge algorithms that localize and quantify cellular features, such as RNA transcripts in single cells [52]. This results in spatially annotated data that can be used by other software programs for single-cell analysis and precise spatial genomic mapping. The NanoString Cosmx is a spatial multiomics single-cell imaging solution used to profile gene and protein expression from FFPE tissues at single-cell and subcellular resolution [51]. The applications of these technologies for HCC research are promising as the tissue transcriptomics can be divided into various neighborhood clusters that are able to phenotype the tissue microenvironment in unprecedented detail.

For comparison, novel technologies such as the dual-aptamer activated proximity-induced droplet digital-PCR can quantify tumor-derived exosomal proteins [53]. This approach can provide valuable information on the role that TAMs play in the TME since HCC cells release exosomes, such as miR-23a-3p, which upregulates PDL-1 expression in macrophages [54]. This has clinical implications, as the expression of PDL-1 is a known predictor of response to immune checkpoint inhibitors [55]. Similarly, the sensitivity of digital droplet PCR and next generation sequencing can detect expression of biomarkers that are of low concentration, and they have shown greater diagnostic efficacy than more conventional methods such as RT-PCR [56]. Although these techniques are able to provide genomic information, they lose critical information about spatial context. Robust analysis of TAM phenotypes in the TME must include spatial relationships with neighboring cells. As mentioned, the crosstalk and proximity of different cell populations is key to understanding TME biology. 

### 4.3. Digital Spatial Profiling of Biomarkers and Fetal-like TAMs

DSP techniques have been used in an elegant study to more closely examine the association between the onco-fetal reprogramming of endothelial cells and TAMs in HCC. In the study, Sharma et al. [57] investigated the link between phenotypic features of malignant cells and early development programs. They observed that the HCC tissue microenvironment was similar to the ecosystem of fetal development, expressing fetal-associated endothelial cells and fetal-like TAMs. The biological implication of this observation is that malignant cells take advantage of fetal programs to generate a fetal microenvironment that is immunologically more tolerogenic. In this fetal microenvironment, tumor cells evade the immune system, which is more tolerogenic than the adult microenvironment. Also, the identification of fetal TAMs agrees with the prevailing view that, within the tissue microenvironment, the behavior of each TAM is not biologically equivalent. The authors proposed that these fetal programs are potential targets for therapy and may yield biomarkers to stratify patients into immune checkpoint inhibitor responders and non-responders. 

### 4.4. Digital Spatial Profiling Limitations and Future Directions

Although these platforms have already provided new insights into the role of TAMs in HCC, GeoMx DSP and Visium are not without limitations, which include: (1) the data are restricted to a ROI, (2) not all genes and proteins are available, (3) the mechanism cannot be confirmed or induced by DSP because this platform is not a functional test, (4) the WTA and protein essay are performed separately, on two different slide sections from a tissue biopsy, (5) high complexity, (6) cost, (7) requires at least 150–200 crlls/ROIs for sufficient counts, (8) ROIs limited to an area of illumination of 600 μm in diameter, and (9) deeper sequencing is required if rare targets are need to be identified, which is more costly and time consuming. In addition, GeoMx DSP and Visium are currently limited for use in human or mouse tissues only.

## 5. Single-Cell RNA Sequencing

### 5.1. Single-Cell RNA Sequencing: Genomic Characterization of Individual Cells

The development of single-cell RNA sequencing (scRNA-seq) started with single-cell quantitative PCR (qPCR) [58]. The latter technique quickly advanced into a whole transcriptome analysis using microarrays. The adaptation to single-cell RNA sequencing followed, and the first scRNA-seq results were published in 2009 [59]. The technique was hastily adopted as the single cell resolution offers new insights into cell biology; it is more sensitive and can detect 75% more genes than microarrays [59]. 

### 5.2. Single-Cell RNA Sequencing Workflow

As shown in Figure 4, the technique consists of isolating single cells for lysis. The RNA molecules of interest are purified and extracted from the total single-cell RNA. Using a reverse transcription reaction, single-stranded RNA is converted into complementary DNA. Subsequently, RNA-seq libraries are created by adding adapters and barcodes that are sequenced using a next generation sequencing (NGS) platform. The data generated by the NGS are trimmed, filtered, and analyzed by a computer software algorithm. Finally, this analysis identifies specific phenotypes and subpopulations of rare cell types. However, unlike the previous platforms discussed, the tissue architecture is not preserved.

### 5.3. Decoding the Tumor Microenvironment of Hepatocellular Carcinoma One Cell at a Time

More studies using scRNA-seq are likely to reveal crucial aspects of the TME and TAM phenotypes, such as population heterogeneity and changes associated with chemotherapy, prognosis, and tumor progression. The tissue microenvironment of HCC is complex, as it is composed of multiple cellular phenotypes, such as TAMs, tumor infiltrating lymphocytes, stromal cells, malignant cells, cytokines and chemokines, and physical barriers, such as the extracellular matrix [60]. Single-cell resolution has revealed new paradigms in the crosstalk between macrophages and the microenvironment. For instance, ScRNA-seq was used to describe how some microenvironmental cues determine macrophage polarization and the fate of tissue fibrosis [61]. Another research group observed in a single-cell study, how transcriptional profiling of bone marrow macrophages identified which subpopulation was activated [62]. Furthermore, single-cell studies have shown that within the same tissue, subsets of immune cells respond differently to the same stimulus. Another study analyzed the immune system by using biopsies from a cohort of eight patients with HBV-associated HCC. They found a negative correlation between the proportion of M2 TAMs and the proportion of tumor-infiltrating T cells in scRNA-seq and deconvoluted bulk-cell RNA-seq datasets [60]. This result was confirmed by IHC of CD8^+^ T cells and CD163^+^ M2 TAMs in HCC biopsy specimens [60]. They also observed that expression of the immunosuppressive marker LAIR1 was enriched in TAM cell clusters expressing the cancer promoting M2 macrophage marker CD163 [60]. As a result of using scRNA-seq, certain TAM characteristics have been identified, which are potential therapeutic targets of immune checkpoint inhibitors.

### 5.4. Single Cell RNA Sequencing Limitations and Future Directions

A limitation of using scRNA-seq is the loss of tissue architecture because it requires tissue dissociation. This hinders a robust spatial analysis of the tissue microenvironment cell-cell interactions. ScRNA-seq allows for the pre-selection of a unique set of genes to study. This pre-selection of genes is restricted to the availability of primers and specific antibodies, as well as to the limited number of genes and proteins of interest. This decreases the amount of genomic data that can be extracted from a tissue because the tissue available to study is scarce.

The experimental design and research question being addressed should be considered before selecting a scRNA-seq platform. Each scRNA-seq platform is unique in the following common steps: (1) cell isolation method (e.g., FACS vs. microfluidics), (2) number of cells required per experiment, (3) cost per cell analysis, and (4) the sensitivity and the amount of genes detected per cell for cell lines and per cell for primary cells is different [62]. Investigators must plan carefully before choosing which scRNA-seq platform is suitable for their research study.

## 6. Conclusions

This review has demonstrated that several cutting-edge platforms are unveiling new paradigms in the role TAMs play in the tissue microenvironment of HCC. The advantages and disadvantages of each platform are highlighted in Table 1. Despite their unique methods, there is a common pattern in the experimental data. For example, experiments using the platforms discussed in this review support the critical role TAMs play in the tumor immune microenvironment and progression of HCC [24,41,57,60,63]. Through unique pathways, each TAM phenotype can create an immunosuppressive tissue microenvironment that favors the proliferation of malignant cells [24,41,57,60,63]. This strongly suggests that no single TAM-targeted therapy will be effective in every HCC case and that more than one phenotype of TAM may have to be targeted in each patient. Two important questions that remain unanswered are whether M2 macrophages favor the progression of the tumor because they become dysfunctional, or because malignant cells take advantage of the immunosuppressive microenvironment created by M2 macrophages. These questions are crucial because they will provide substantial evidence to support the use of macrophage targeted therapy as an adjuvant therapy in the treatment of HCC. 

These cutting-edge platforms are being used more commonly in the cancer field due to increased use of immunotherapy regimens and more clinical desire to predict a patient’s response to therapy before initiation. The addition of immune checkpoint inhibitors, as adjuvant treatment for different types of tumors, including HCC, is becoming more common and the efficacy of immune checkpoint inhibitors, depends on the immune cell topology of the tissue microenvironment [64,65]. Soon, the presence of unique phenotypes of immune and stromal cells will determine if immunotherapy is clinically indicated in HCC. Immune checkpoint inhibitors are currently used as a second line therapy for patients with HCC who are non-responsive to first line sorafenib or have advanced disease [66]. Furthermore, recent data indicates that, in addition to T cells, other cells, and components of the tissue microenvironment, such as B cells, myeloid lineage cells, cancer-associated fibroblasts, and the vasculature play critical roles in the effectiveness of immune checkpoint blockade [64]. Successful personalization of immunotherapy requires accurate characterization of the tissue microenvironment. Techniques that may be subtly changing the phenotype of these cellular targets during isolation, such as flow cytometry, may be missing some key components to advance our knowledge of TAM biology.

In HCC, response to immunotherapy is evaluated by using different classification systems, radiologic images, clinical parameters, and tissue biopsy, which is the gold standard. Although there is a clinical trend towards thinking that non-invasive testing is equal or superior to invasive testing, it should be explicitly mentioned that non-invasive testing loses exponential amounts of information about the tissue architecture. In a tissue biopsy the submicron resolution of the spatial context is 10,000 times more than the standard MRI (40× whole slide 0.11 × 0.11 μm^3^ vs. MRI 1.5 × 1.5 mm^3^). A study of Hodgkin’s lymphoma patients, showed that MSI and digital imaging analysis are effective strategies of profiling the immunomodulatory proteins expressed by multiple populations of immune cells in the tissue microenvironment, which may guide clinicians’ choice of immunotherapy or combination therapy [65]. In addition, despite years of research, systemic biomarkers that can provide diagnostic and/or prognostic information remain limited in HCC [67,68]. 

In summary, regardless which of the above platforms are utilized, there are clear data supporting the following: (1) TAMs may be negative regulators of the immune response in the TME, (2) clinical outcomes may be impacted by the type of TAM present in the tissue microenvironment, (3) TAMs can decrease the efficacy of immune checkpoint inhibitors, and (4) targeted therapy against TAMs is a promising avenue for personalized precision medicine approaches for the treatment of HCC [24,69].

## Figures and Tables

**Figure 1 cancers-14-01861-f001:**
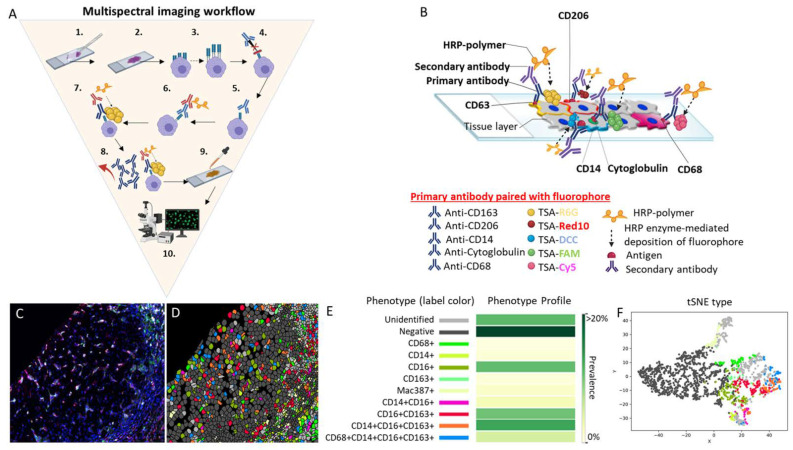
Multispectral imaging workflow. (**A**) MSI consists of the following 10 steps: (1) deparaffinization of the tissue slide, (2) slide fixation, (3) antigen retrieval, (4) addition of a blocking reagent to prevent non-specific binding of antibodies to tissue or to Fc receptors, (5) incubation and binding of an unmodified primary antibody to its target, (6) HRP-conjugated secondary antibody incubation and binding of the HRP-conjugated antibody to the primary antibody, (7) HRP enzyme-mediated in situ deposition of the tyramine-fluorophore that covalently binds to the tissue near the target, (8) antibody removal, (9) counterstaining of the tissue with DAPI, and (10) mounting of the slide and visualization using fluorescent microscopy. Note that, after step 8, steps 3 to 8 are repeated one to four additional times or until the maximum number of antibodies (five for multiplex) is reached. (**B**) This is a representative image of a 5-plex stained slide with the membrane antibodies CD163 paired with the fluorophore TSA-Rhodamine 6G (gold), CD206 paired with the fluorophore TSA-Red 10 (red), and CD14 paired with the fluorophore TSA-DCC 10 (aqua blue). A nuclear marker is represented by the antibody anti-cytoglobulin paired with the fluorophore TSA-FAM (green). The cytoplasmic marker is represented by CD68 paired with the fluorophore TSA-Cy5 (violet). In the figure, all five antibodies are shown binding to the tissue at the same time to illustrate how different fluorophores signals are produce simultaneously in situ. However, the antibodies do not remain bound to the tissue after step 8. Only the fluorophores are attached to the tissue after step 8, as they form a covalent bond in the vicinity where the antibody was previously bound to the antigen, amplifying the signal that is acquired by the IF microscope. (**C**) We stained a representative human liver biopsy slide from a patient with NASH and advanced fibrosis with a multiplex macrophage panel: CD68 (green-Opal 520), CD14 (Magenta-Opal 540), CD16 (red-Opal 620), CD163 (cyan-Opal 650), MAC387 (white-Opal 690), and nuclear stain (Blue-DAPI). A single 20× fluorescence image was obtained after spectral unmixing was applied. (**D**) Single multiplex images from Figure 1C were used to export multicomponent TIFF files for Visiopharm analysis. A phenotyping application was used to determine the number of different cellular phenotypes present in each image. Each colored dot represents a unique cellular phenotype. Dark gray dots represent cells that were negative for all the markers in the multiplex panel. (**E**) The same exported multicomponent TIFF files used in Figure 1D were used by the Visiopharm phenotype matrix algorithms to determine the different cellular phenotypes present in each of the 20× multiplex images. The colors shown for each of the different phenotypes correlate with the various colored dots (i.e., individual cells) shown in the multiplex image from Figure 1D. Dark green and white boxes indicate the populations with the highest and lowest prevalence, respectively. (**F**) t-SNE plots highlight the unique patterns of concatenated cellular markers that are present in the image. This algorithm uses nonlinear dimensional reduction to allow visualization of high dimensional data sets. Cells with similar properties appear closer together and those that are dissimilar appear farther apart in the 2-dimensional map. Abbreviations: 1^st^ Ab, primary antibody; 2^nd^ Ab, secondary antibody; Ab, antibody; DAPI, 4′,6-diamidino-2-phenylindole; HRP, horseradish peroxidase; TSA, tyramine signal amplifier.

**Figure 2 cancers-14-01861-f002:**
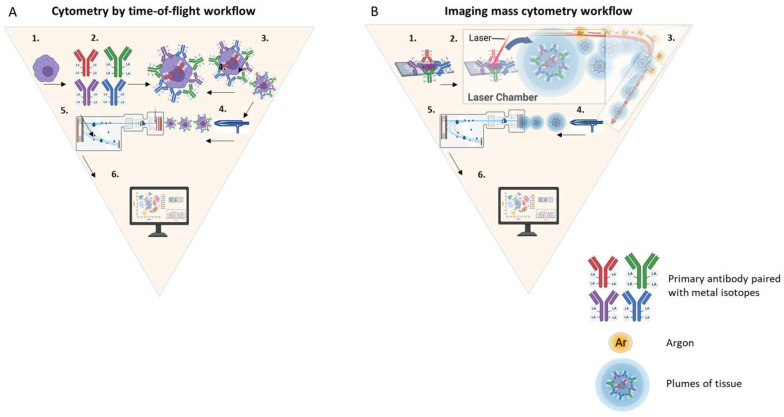
Cytometry by time-of-flight workflow. (**A**) Suspension CyTOF consists of the following steps: (1) cells are selected, (2) antibodies are conjugated with a transition element isotope, (3) the cells are incubated with the antibodies conjugated with a transition element isotope, (4) the cells with an epitope-specific antibody conjugated to a transition element isotope reporter are vaporized, (5) vaporized cells enter the time-of-flight analyzer, where the isotopes are quantified, and (6) data are analyzed by a computer software. (**B**) Imaging mass cytometry consists of the following steps: (1) tissue is labeled with a pre-selected panel of antibodies conjugated with stable isotopes, (2) the tissue is inserted into an ablation chamber and a camera (not shown) acquires images from the tissue slide, (3) the laser beam ablates a spot of tissue stained with the metal-tagged antibodies, forming aerosol plumes, (4) plumes of tissue matter are directed by a flow of inert gas (Argon) into a coupling tube that delivers them to an inductively coupled plasma ion source, where they are vaporized, atomized, and ionized, (5) the metal-isotope ion cloud passes into the TOF mass spectrometer for analysis of isotope abundance, and (6) data are analyzed using computer software. Abbreviations: Ab, antibody; Ar, argon.

**Figure 3 cancers-14-01861-f003:**
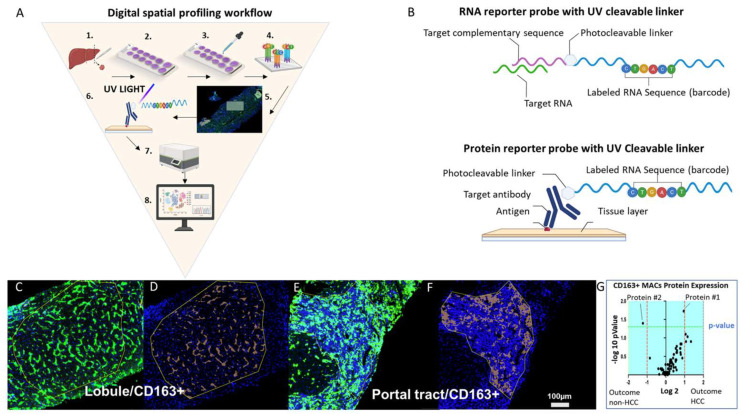
Digital spatial profiling workflow (**A**) The workflow involves these steps: (1) tissue of interest is biopsied (e.g., liver), (2) tissue is sectioned and fixed to a positively charged glass slide, (3) the slide is stained with up to four morphology markers and a nuclear stain, (4) probes are incubated with the tissue; (5) regions of interest (ROIs) are selected after incubation; (6) target-specific barcodes are liberated by the ultraviolet laser from the selected ROIs, (7) targets are quantified by the nCounter platform, and (8) data are analyzed using a combination of imaging analysis software and R scripts, several of which are provided by the company. (**B**) The top image represents an RNA probe that is composed of a target complementary sequence conjugated with a photocleavable linker that bonds oligonucleotide RNA sequence barcode. The bottom image represents a protein probe that is composed of a target antibody conjugated with a photocleavable linker that bonds an oligonucleotide RNA sequence barcode. (**C**) Representative image of a ROI within the liver lobule and (**D**) represents the respective CD163^+^ tissue segmentation. (**E**) Representative image of a ROI within the portal tract and (**F**) represents the respective CD163^+^ tissue segmentation. (**G**) Example of a volcano plot showing enriched liver protein expression by CD163^+^ macrophages in the liver lobule of HCV+ patients with cirrhosis who did and did not develop HCC. The plot was generated by probing protein expression from a liver biopsy. Abbreviations: Ab, antibody; ROI, region of interest; UV, ultraviolet. Scale bar = 100 µm.

**Figure 4 cancers-14-01861-f004:**
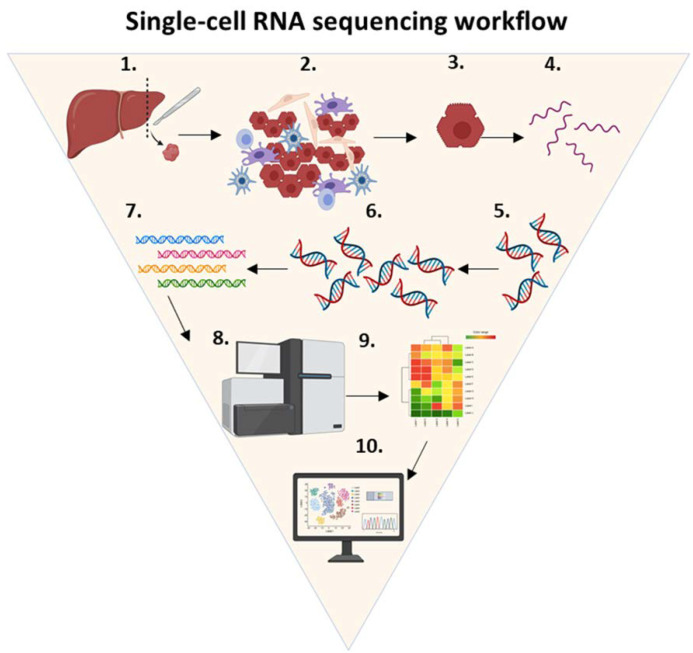
Single-cell RNA sequencing workflow. ScRNA-seq is a technique that consists of the following steps: (1) tissue of interest is selected (e.g., liver), (2) the tissue is dissociated using enzymatic digestion, (3) isolated single cells are lysed, (4) RNA molecules of interest are purified and extracted from the total single-cell RNA, (5) a reverse transcription reaction is used to convert single-stranded RNA into complementary DNA, (6) complementary DNA is amplified using real-time polymerase chain reaction, (7) RNA-seq libraries are created by adding adapters and barcodes, (8) DNA is sequenced using a next generation sequencing platform, (9) the data generated by the NGS are trimmed, filtered and analyzed by computer software algorithms to yield single-cell expression profiles, and (10) cells are mapped to specific phenotypes and subpopulations of rare cell types. Abbreviations: cDNA, complementary DNA.

**Table 1 cancers-14-01861-t001:** Summary of strengths, weakness, opportunities, and threats for each platform.

Platform	Strengths	Weakness	Opportunities	Threats
MSI	Provides in situ visualization of multiple cell phenotypes, while preserving tissue architecture Data can be used for spatial and nearest-neighbor analyses Allows data collection across entire tissue section	Limited amount of markers can be analyzed Spectral overlap can hinder colocalization analysis and biomarker quantification	Presence of specific TAMs in the TME may allowed for personalized treatment Automated equipment for high throughput is available Can be incorporated into routine brightfield microscopy [10]	Additional standardization is needed prior to clinical implementation [11,27] Most pathologist lack training with MSI microscopy [11]
IMC	Allows for in situ visualization of more than 40 targets [35,39] No spectral overlapping The use of non-biologic markers increases signal-to-noise ratio [38]	Vaporization of cells and tissue ablation are irreversible steps	Provides high-dimensional analysis of various cellular features [35] Allows for stratification of patients into treatment responders and non-responders [41,47]	Targets with low exppression or specific populations of cells may be challenging to detect [35,48] Complexity of data analysis [35,48]
DSP	Spatial characterization of both RNA and protein expression using a limited amount of tissue Allows for morphological segmentation of a unique population of cells [57] Combines high-plex microscopy and spatial genomics	Analysis is limited to pre-selected proteins and RNA probes Gene and protein assays are evaluated on two sequential slide sections from a tissue block	Gene expression and protein profiling in a single ROI Single cell resolution is in development	High Cost Requires at least 150–200 cells per ROI for sufficient counts Rare target analysis is more costly and time consuming
ScRNA-seq	Single cell RNA resolution transcriptomics Accurate identification of specific phenotypes and subpopulations of rare cell types	Loss of tissue architecture [62] No spatial co-localization analysis	Characterization of cellular crosstalk at single cell resolution Profile potential therapeutic targets for rare phenotypes [60,61]	Cell isolation method, number of cells per experiment, cost per cell and sensitivity vary between scRNA-seq platforms [62]

Abbreviations: DSP, digital spatial profiling; IMC; imaging mass cytometry; MSI, multispectral imaging; ScRNA-seq, single-cell RNA sequencing; TAMs, tumor-associate macrophages.

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
