# Peer review of "Cutting-Edge Platforms for Analysis of Immune Cells in the Hepatic Microenvironment—Focus on Tumor-Associated Macrophages in Hepatocellular Carcinoma"

_cancers, 2022, doi:10.3390/cancers14081861_

Round 1

Reviewer 1 Report

The role of tumor-associated macrophages (TAM) in the pathogenesis of hepatocellular carcinoma (HCC) is poorly understood. Fortunately, many platforms for the analysis of immune cells in the hepatic microenvironment are currently being developed.

This review describes the workflow of each platform, summarizes recent research using these approaches and explains the advantages and limitations of each by targeting the study of the role of tumor-associated macrophages in hepatocellular carcinoma.

This review is informative, clear and well structured.

The conclusion could have benefited from a SWOT-style summary diagram/table (Strengths - Weaknesses - Opportunities - Threats).

Other platforms are in development and they could also have been mentioned:

-nanoString CosMx

- Miltenyi MACSIma

- Rebus ESPER

- Leica Microsystems Cell Dive

- RareCyte Orion

Author Response

Reviewer # 1:

The role of tumor-associated macrophages (TAM) in the pathogenesis of hepatocellular carcinoma (HCC) is poorly understood. Fortunately, many platforms for the analysis of immune cells in the hepatic microenvironment are currently being developed.

This review describes the workflow of each platform, summarizes recent research using these approaches and explains the advantages and limitations of each by targeting the study of the role of tumor-associated macrophages in hepatocellular carcinoma.

This review is informative, clear and well structured.

  1. The conclusion could have benefited from a SWOT-style summary diagram/table (Strengths - Weaknesses - Opportunities - Threats).

Author’s Response:

Thank you for your positive comments and great suggestions.  We agree that the conclusion could benefit from a SWOT-style summary diagram/table. Therefore, we have included a SWOT-style summary table (See page. 17), which summarized the main strengths, weakness, opportunities and threats of each of the platforms described in the review. We hope this table will help the reader choose the most suitable approach for their research needs.

  1. Other platforms are in development and they could also have been mentioned:

-nanoString CosMx

- Miltenyi MACSIma

- Rebus ESPER

- Leica Microsystems Cell Dive

- RareCyte Orion

Author’s Response:

As suggested, we mentioned and highlighted the main features of Miltenyi MACSIma, Leica Microsystems Cell Dive, and RareCyte Orion platforms in our MSI cyclic technique description (See page 5; PMID: 33868223, PMID: 34732839, PMID: 35308236). The nanoString CosMx and Rebus ESPER techniques are included in our Digital Spatial Profiling discussion (See page 11 and 12; DOI: 10.1101/2021.11.03.467020; PMID: 34616070). However, these state-of-the-art platforms are under development and have not been widely used in the study of macrophages in HCC.

Reviewer 2 Report

I think that some parts can be reduced in content as they are too detailed in the description of the sample preparation, but overall the work is well written and well structured.
I suggest to check the references and to use the same style for all the citations.

Author Response

Reviewer 2:

  1. I think that some parts can be reduced in content as they are too detailed in the description of the sample preparation, but overall the work is well written and well structured.

Author’s Response:

Thank you for your comments. We included a detailed description of each technique to help investigators to choose the most suitable platform for their research needs.

  1. I suggest to check the references and to use the same style for all the citations.

Author’s Response:

As suggested, we used the MDPI endnote style reference format recommended by the journal.

Reviewer 3 Report

Certainly a very interesting review of the literature concerning modern approaches to the study of the immunophenotype of cells, including tumor associated macrophages. Of particular interest is an approach that allows assessing the immunophenotype of macrophages and their microenvironment without isolating them from the liver, which eliminates the aggressive effects of enzymes and mechanical disaggregation.

Despite the presence of merits, I would like to make some comments to the authors.

Despite the fact that the main purpose of the article is to review modern research methods, the title still focuses on macrophages associated with liver cancer. In this regard, it seems to me that the review of the literature should be supplemented with modern data on the immunophenotype of  Hepatocellular Carcinoma associated macrophages, as well as on the available therapeutic approaches that consider liver macrophages as possible therapeutic targets.

Author Response

Reviewer # 3

Certainly a very interesting review of the literature concerning modern approaches to the study of the immunophenotype of cells, including tumor associated macrophages. Of particular interest is an approach that allows assessing the immunophenotype of macrophages and their microenvironment without isolating them from the liver, which eliminates the aggressive effects of enzymes and mechanical disaggregation.

Despite the presence of merits, I would like to make some comments to the authors.

  1. Despite the fact that the main purpose of the article is to review modern research methods, the title still focuses on macrophages associated with liver cancer. In this regard, it seems to me that the review of the literature should be supplemented with modern data on the immunophenotype of Hepatocellular Carcinoma associated macrophages, as well as on the available therapeutic approaches that consider liver macrophages as possible therapeutic targets.

Author’s Response:

Thank you for positive comments.  As you mentioned, the main purpose of this review is to describe the state-of-the-art platforms that have recently emerged, highlight their advantages and disadvantages, and present HCC studies where they have been used. We have cited several studies where these platforms were useful to characterize the tumor-associated macrophages (TAMs) in the hepatic microenvironment (See ref PMID: 34253573, PMID: 32976798, PMID: 28514441, PMID: 34140495). We have added 2 new references (PMDI 33815418; PMID 33619259). We also mentioned methodologies used to identify the enrichment of macrophage markers that could be used as potential therapeutic targets for the treatment of HCC (See page 9,12,14; PMID: 34253573, PMID: 33680573, PMID: 34140495, PMID: 32976798).

Reviewer 4 Report

Manuscript is a well written.

  1. Authors may add the section of digital droplet PCR and next-generation sequencing application for tumor associated macrophage.

See the following references:

Della Starza I, et al. Digital droplet PCR and next-generation sequencing refine minimal residual disease monitoring in acute lymphoblastic leukemia. Leuk Lymphoma. 2019 Nov;60(11):2838-2840. doi: 10.1080/10428194.2019.1607325. PMID: 31050551

Lin B, et al. Tracing Tumor-Derived Exosomal PD-L1 by Dual-Aptamer Activated Proximity-Induced Droplet Digital PCR. Angew Chem Int Ed Engl. 2021 Mar 29;60(14):7582-7586. doi: 10.1002/anie.202015628. Epub 2021 Feb 25. PMID: 33382182

Author Response

Reviewer # 4

Manuscript is a well written.

  1. Authors may add the section of digital droplet PCR and next-generation sequencing application for tumor associated macrophage.

See the following references:

Della Starza I, et al. Digital droplet PCR and next-generation sequencing refine minimal residual disease monitoring in acute lymphoblastic leukemia. Leuk Lymphoma. 2019 Nov;60(11):2838-2840. doi: 10.1080/10428194.2019.1607325. PMID: 31050551

Lin B, et al. Tracing Tumor-Derived Exosomal PD-L1 by Dual-Aptamer Activated Proximity-Induced Droplet Digital PCR. Angew Chem Int Ed Engl. 2021 Mar 29;60(14):7582-7586. doi: 10.1002/anie.202015628. Epub 2021 Feb 25. PMID: 33382182

Author’s Response:

Thank you for your comments and feedback. We agree that these techniques have the potential to provide valuable information on the role tumor-associated macrophages (TAMs) play in the tumor microenvironment. (See page 12, PMID:33680573). The ddPCR and NGS platforms are important for detecting the expression of low concentration biomarkers, but critical information about the architecture and spatial context is limited. For this reason, we decided to highlight some important features of these techniques in our digital spatial profiling discussion without adding a section describing these platforms (See page 12).

Round 2

Reviewer 3 Report

All comments were answered satisfactorily.